# How face perception unfolds over time

Katharina Dobs [1,2,3], Leyla Isik [1,2,3], Dimitrios Pantazis [1,2] & Nancy Kanwisher[1,2,3]

Within a fraction of a second of viewing a face, we have already determined its gender, age and identity. A full understanding of this remarkable feat will require a characterization of the computational steps it entails, along with the representations extracted at each. Here, we used magnetoencephalography (MEG) to measure the time course of neural responses to faces, thereby addressing two fundamental questions about how face processing unfolds over time. First, using representational similarity analysis, we found that facial gender and age information emerged before identity information, suggesting a coarse-to-fine processing of face dimensions. Second, identity and gender representations of familiar faces were enhanced very early on, suggesting that the behavioral benefit for familiar faces results from tuning of early feed-forward processing mechanisms. These findings start to reveal the time course of face processing in humans, and provide powerful new constraints on computational theories of face perception.

[1] Department of Brain and Cognitive Sciences, Massachusetts Institute of Technology, Cambridge, MA 02139, USA. [2] McGovern Institute of Brain Research, Massachusetts Institute of Technology, Cambridge, MA 02139, USA. [3] The Center for Brains, Minds, and Machines, Massachusetts Institute of Technology, Cambridge, MA 02139, USA. Correspondence and requests for materials should be addressed to K.D. (email: katharina.dobs@gmail.com)

A brief glimpse at a face quickly reveals rich multi-dimensional information about the person in front of us. How is this impressive computational feat accomplished? A key property of a complex computation is that it proceeds via stages and hence unfolds over time. Thus, knowing which information is extracted when is of the essence for understanding the computations underlying face perception. Surprisingly, it remains unknown 1) when after stimulus onset information about different face dimensions, such as gender, age, or identity, are extracted, and 2) how early these computations are affected by the familiarity of the face. Here we use magnetoencephalography (MEG) to answer these questions.

While extensive prior evidence indicates that humans detect and recognize faces very rapidly[1–3], much less is known about the precise temporal dynamics of extraction of information about different dimensions of face information. One possibility is that different dimensions of face information are extracted at different stages of processing. For example, gender information might be extracted before identity information, following a coarse-to-fine trajectory[1,4]. Alternatively, different face dimensions could be processed at the same time, suggesting greater interdependence of their processing[5,6]. Resolving the time course by which information about gender, age, and identity emerge will importantly constrain computational models of face perception.

Our second question is whether these face dimensions are processed differently for familiar vs. unfamiliar faces, and if so how early familiarity affects processing. A striking yet unexplained finding about face perception is that familiar faces are processed more robustly and efficiently than unfamiliar faces[7]. The neural mechanisms underlying this effect remain unknown. According to one hypothesis, visual experience with specific faces tunes the bottom-up processing filters for face features, thereby enhancing representations of familiar faces[8]. This hypothesis predicts that familiarity should enhance face representations early in processing, at the same time when those representations are first being extracted. Alternatively (or in addition[9]), familiarity effects in face processing could arise via activation of associated person knowledge and memories, which would then enhance perceptual representations in a top-down manner[10,11]. This hypothesis predicts that familiarity should enhance face representations at some point after those representations are first extracted. Determining which (or both) of these accounts is correct will provide an important step towards understanding the neural mechanisms underlying the behavioral familiarity enhancement effect, and will further inform more general and long-standing questions of how specific prior experience affects the processing of objects[12–14].

To determine how face processing unfolds over time, we applied multivariate analysis methods to MEG data from subjects viewing images of familiar and unfamiliar celebrities who varied orthogonally in gender and age. We used representational similarity analysis (RSA) to reveal the temporal dynamics of the representation of gender, age, and identity for familiar and unfamiliar faces. We show two novel results: 1) the brain encodes gender and age information before identity information, and 2) information about both identity and gender is enhanced for familiar faces early on in processing. These findings constrain computational models of face perception and support a bottom-up account of the strong familiarity effects previously reported in behavior.

## Results

**Behavioral performance during MEG task**. We recorded MEG data from 16 subjects while they viewed five images of each of eight familiar (American) and eight unfamiliar (German) celebrities and monitored for consecutive repetitions of identical images (i.e., 1-back task; Fig. 1a). Celebrities varied orthogonally in gender and age. Images for each identity were chosen to incorporate natural variability in various dimensions, such as pose, hair style, eye gaze or lightning. Subjects viewed 28 repetitions of each of 80 face stimuli, each presented for 200 ms in individual trials. To ensure that subjects maintained attention and processed the presented images throughout the course of the experiment, we asked them to perform a 1-back task, pressing a button when the identical image was repeated consecutively. Subjects were highly sensitive to an image repetition (mean sensitivity index d' ± SEM: 4.28 ± 0.16) and responded quickly (mean response time ± SEM: 458 ± 10 ms after target stimulus onset). To test whether subjects processed familiar faces more efficiently than unfamiliar faces, we compared the responses to familiar versus unfamiliar images. While subjects' sensitivity did not differ between familiar and unfamiliar face images ($p = 0.67$; two-sided signed-rank test), subjects responded significantly faster to familiar than to unfamiliar faces ($p = 0.021$; familiar: 454 ± 11 ms, unfamiliar: 462 ± 9 ms). These behavioral results confirm that the processing of familiar faces is enhanced compared to unfamiliar faces, even when the task requires only image-level (not identity-level) processing.

**Time course of face image decoding**. To reveal the time course of face processing, we performed multivariate pattern analysis on the MEG signals in a time-resolved manner (Fig. 1b)[15,16]. We first extracted a set of principal components (PCs) for each subject, based on MEG responses across sensors, trials, timepoints (−100 to 800 ms with respect to image onset; 1 ms resolution) and conditions. Then, using the resulting PCs, we trained and tested support vector machines (SVM) on every pair of face stimuli for each time point. Dissimilarity for each pair of stimuli was computed as five-fold cross-validated decoding accuracy, resulting in one 80 × 80 MEG representational dissimilarity matrix (RDM) per subject and time point.

To determine when neural representations can first discriminate any visual information, we computed the average across all pairwise decoding accuracy values, separately at each time point. This analysis yielded a time course of neural image decoding accuracy (Fig. 2a). Individual images could be discriminated by visual representations early (decoding first reached significance at 46 ms), reached a peak at 103 ms (73.3% mean decoding accuracy) and remained significantly above chance until 706 ms after stimulus onset (cluster-corrected sign permutation test, cluster-defining threshold $p < 0.05$, corrected significance level $p < 0.05$). This time course is highly similar to the course of image decoding reported in previous MEG studies[15–17] and shows how neural responses are resolved at the level of individual face images. To test how persistent neural responses to face image representations were, we performed temporal generalization analysis (see Supplementary Note 4).

**Early representations of face dimensions revealed by RSA**. To determine when neural representations discriminated face dimensions at higher categorization levels (e.g., gender) or even at the level of identity or familiarity, we created a model RDM (e.g., 1 for between and 0 for within gender stimulus pairs) for every face dimension (i.e., gender, age, identity and familiarity; Fig. 1c). Because some face dimensions (e.g., gender) might be associated with differences in low-level image properties (e.g., long hair versus short hair for female and male, respectively), we further created a low-level feature RDM based on an early layer of a deep, convolutional neural network trained on face identity (i.e., VGG-Face; see Methods for details and for comparison to other low-

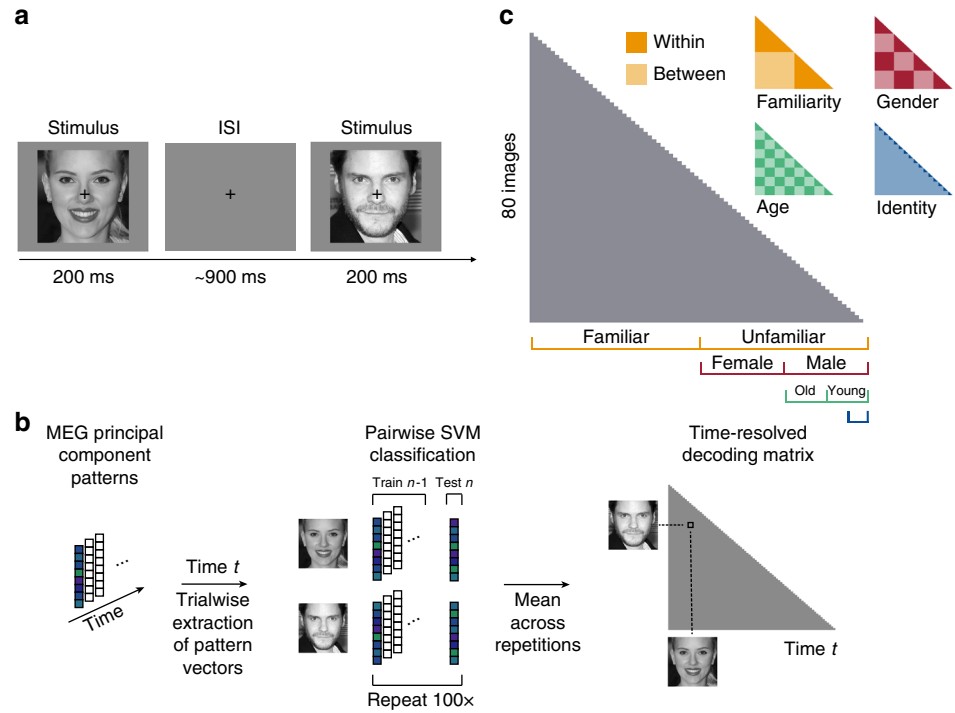

**Fig. 1** Task and multivariate MEG analyses. **a** Subjects viewed 80 images of faces while performing a 1-back task on the image. Each image was presented for 200 ms, followed by a variable [800–1000 ms] interstimulus interval (ISI). **b** MEG analyses were performed in a time-resolved manner on principal components (PCs) extracted from all MEG sensors, separately for each subject (see Methods section). For each time point *t*, we extracted the pattern of response across PCs for each condition and each trial and performed pairwise cross-validated SVM classification. The resulting decoding accuracy values resulted in an 80 × 80 representational dissimilarity matrix (RDM) for each time point. **c** To perform representational similarity analysis (RSA), we constructed model RDMs for each face dimension (1 corresponding to between and 0 corresponding to within category, respectively). Images shown are not examples of the original stimulus set due to copyright; the exact stimulus set is available at [https://osf.io/gk6f5/]. Images shown are in public domain and available at [https://commons.wikimedia.org]

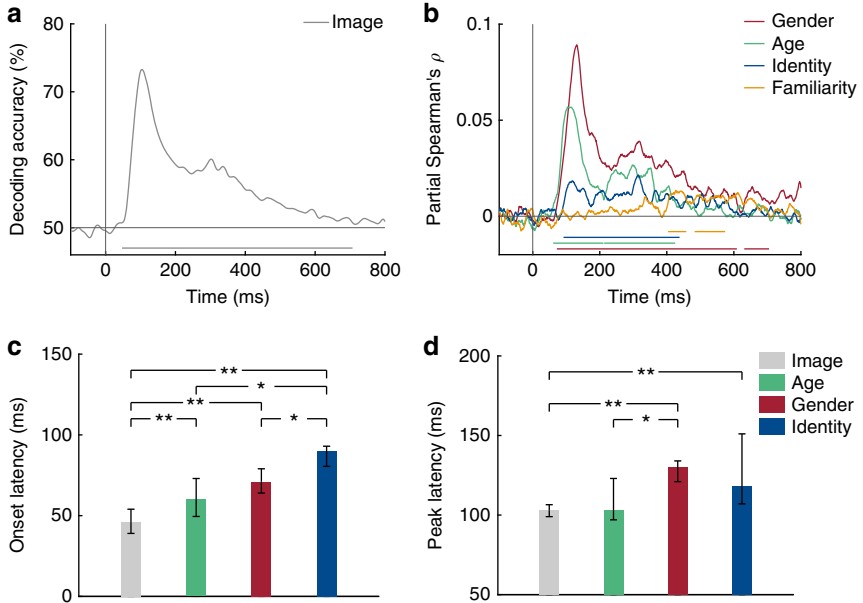

**Fig. 2** Decoding of face images and dimensions from MEG signals. **a** Time course of image decoding where 0 indicates image onset (*n* = 16). **b** Time course of partial Spearman correlations between MEG RDMs and model RDMs (see Fig. 1c) for gender (red), age (green), identity (blue) and familiarity (orange), partialling out all other models and low-level features (see Methods). Lines below plots indicate significant times using cluster-based sign permutation test (cluster-defining threshold $p < 0.05$, corrected significance level $p < 0.05$). **c, d** Onset (**c**) and peak (**d**) latencies for decoding of images, gender, age and identity. Error bars indicate bootstrapped 95% confidence intervals. Stars above bars indicate significant differences across conditions (one-sample two-sided bootstrap test, **$p < 0.01$; *$p < 0.05$; FDR-corrected). Source data are provided as a Source Data file

level feature models). For every face dimension, we computed partial Spearman correlations between corresponding model and MEG RDMs at every time point and for each subject, while partialling out all other model RDMs and the low-level feature model (Fig. 2b). We further performed a similar analysis based on cross-decoding of face dimensions (e.g., training gender on some identities, testing gender on the left-out identities), which yielded highly similar results (see Supplementary Note 3).

MEG responses revealed information about age (significant time points: 60–208, 214–368 ms), gender (72–608, 630–705 ms) and specific identity (91–437 ms; all cluster-corrected sign permutation tests, cluster-defining threshold $p < 0.05$, corrected significance level $p < 0.05$). Age and gender information were extracted first from MEG representations, and arose around 20 ms earlier than identity information (Fig. 2c; $p < 0.05$; one-sample two-sided bootstrap test, FDR-corrected). This finding suggests that coarse distinctions between faces in categorical dimensions are extracted before finer distinctions on an identity level; in line with a coarse-to-fine processing of face dimensions[1,4]. Although the later latency of identity information could in principle reflect lower power to detect this smaller effect, this account seems unlikely, as we explain in the familiarity analysis results in the next section. Importantly, age, gender and identity dimensions were discriminated significantly later than individual images ($p < 0.01$; one-sample two-sided bootstrap test, FDR-corrected) suggesting that these facial dimensions emerged early during processing but after low-level feature extraction. Interestingly, we found that neural representation of gender and identity peaked at similar latencies ~125 ms after stimulus onset (Fig. 2d). So while peak latencies often vary for different types of object categories[16] (presumably processed in different locations in the brain), different dimensions of the same category (here faces) can peak at similar times.

We further found that MEG representations separated familiar from unfamiliar identities at much later latencies (403–457, 482–573 ms; cluster-corrected sign permutation test, cluster-defining threshold $p < 0.05$, and corrected significance level $p <$

0.05) than perceptual categories such as gender or age, and after specific identity information is extracted (all $p < 0.01$; one-sample two-sided bootstrap test, FDR-corrected). This finding indicates that a late signature of generic familiarity can be read out from MEG signals, long after the onset of extraction of identity information. The basis of this familiarity signature is not clear, and could reflect the activation of memories associated with a given familiar individual, an emotional response to a familiar face, or a generic familiarity response.

**Familiarity enhances face information at early stages**. Behavioral evidence shows that familiar faces are processed more robustly than unfamiliar faces, but it is unknown how early in processing this occurs. To answer this question, we conducted the same RSA analysis as above, but did so separately for familiar and unfamiliar faces (Fig. 3a). Note that this separation reduces the data available for each analysis fourfold, thus reducing the signal to noise ratio (SNR). Despite this reduction, MEG representations still discriminated familiar faces by gender (Fig. 3b; 71–206, 226–492 ms), age (Fig. 3c; 92–200, 220–271 ms) and identity (Fig. 3d; 96–168, 252–406, 447–516, 519–570 ms; all cluster-corrected sign permutation tests, cluster-defining threshold $p < 0.05$, and corrected significance level $p < 0.05$). Interestingly, the onset latency of identity information for familiar faces was not earlier than that found across all faces, despite the much higher peak correlation for familiar faces. This result suggests that the later onset for identity information than gender and age information found in the previous analysis (Fig. 2c) is unlikely to reflect overall lower sensitivity to identity information.

In contrast to familiar faces, neural representations of unfamiliar faces were less pronounced but could still be discriminated by gender (Fig. 3b; 102–415, 488–560 ms) and age (Fig. 3c; 70–137, 241–397 ms; all cluster-corrected sign permutation tests, cluster-defining threshold $p < 0.05$, and corrected significance level $p < 0.05$), and were no longer discriminable by identity (Fig. 3d). Crucially, we found that the encoding of gender and identity, but not age, was significantly

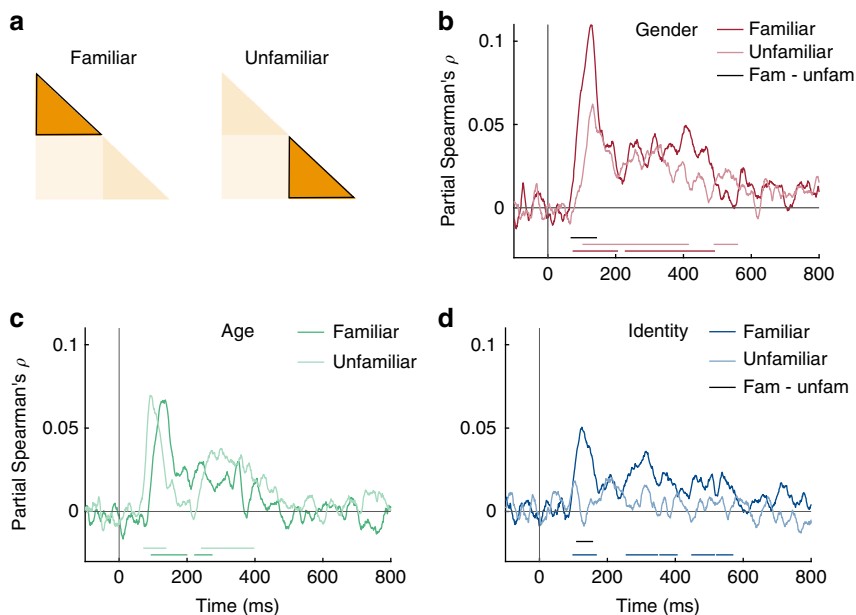

**Fig. 3** Effect of familiarity on face representations. **a** We conducted separate RSA analyses for familiar and unfamiliar faces ($n = 16$). Note that this reduces the amount of data fourfold. **b–d** Time course of partial Spearman correlations between MEG RDMs and gender (**b**), age (**c**) and identity (**d**) separated for familiar and unfamiliar faces, partialling out other models and low-level features (see Methods). Colored lines below plots indicate significant times, and black lines indicate significant difference between conditions both using cluster-based sign permutation test (cluster-defining threshold $p < 0.05$, and corrected significance level $p < 0.05$). Source data are provided as a Source Data file

enhanced for familiar compared to unfamiliar faces (Fig. 3b–d, black lines in plots; all cluster-corrected sign permutation tests, cluster-defining threshold $p < 0.05$, and corrected significance level $p < 0.05$). These enhancements occurred early with respect to the onset of gender and identity encoding (gender: 66 ms – 142 ms (onset 71 ms); identity: 106–159 ms (onset 96 ms)). Overall, familiarity enhancement arose early during processing, suggesting that early stages of visual processing are tuned to familiar face features (perhaps via local recurrent processes[17]).

**MEG responses are correlated with behavior.** Not all information that can be read out from brain activity is used by the brain to guide behavior[18,19]. Is the face information we report here related to behavior? To find out, 14 of the 16 subjects participated in a behavioral multi-arrangement paradigm (see Methods) after the MEG experiment to assess their perceived dissimilarity between the face stimuli. We correlated the resulting behavioral RDM of each subject with the MEG RDMs for each time point for that subject, while partialling out low-level stimulus features. The resulting correlation time courses revealed a significant partial correlation between MEG and behavior (Fig. 4a; 85–274, 294–420 ms after stimulus onset; cluster-corrected sign permutation test, cluster-defining threshold $p < 0.05$, and corrected significance level $p < 0.05$) with a peak at 123 ms (mean partial Spearman's $\rho$: 0.06) after stimulus onset. This peak overlaps with the peak of correlations obtained for gender, age and identity (see Fig. 2d). To assess how much of the explainable MEG variance was captured by behavior, we further computed an estimate of the noise ceiling given the variability across the restricted set of fourteen subjects (gray-shaded area in Fig. 4a). While the correlation between behavior and MEG reveals the shared variance between both modalities, it does not indicate how much each of the face dimensions contributes to this shared variance. To answer this question, we conducted a model-based commonality analysis (Fig. 4b; see Methods). This approach is based on variance partitioning and identifies the variance uniquely shared between MEG and behavior and a given model RDM (e.g., the gender model), termed commonality coefficients. We restricted this analysis to the time window during which we found significant correlations between MEG and behavior (i.e., 85–420 ms after stimulus onset; note the changed x-axis in Fig. 4b). Given the late correlation with the MEG data and the familiarity model (see Fig. 2b), we did not include the familiarity model in this analysis. We found that gender, age and identity each uniquely contributed to the shared variance (commonality coefficients are shown in Fig. 4b). We further show the explained variance

between MEG and behavior as reference (gray line in Fig. 4b). Note that commonality coefficients reported with this kind of analysis are often significant but very small[20]. Together, we found that behavior was predictive for MEG responses and that the shared variance predominantly reflected gender information, followed by age and then identity, though all three were significant.

## Discussion

This study answers two fundamental questions about the time course of face processing in humans. First, we find that extraction of information about gender and age begins after image-level decoding but before extraction of identity information. Second, we show that familiarity of the face enhances representations of gender and identity very early in processing. These two new findings reveal the temporal dynamics underlying face processing and provide powerful constraints on computational models of face perception. Next, we relate these findings to prior work in monkeys and humans, as well as computational models of face perception.

While a few prior studies have investigated the time course of face perception using multivariate pattern analyses in humans[21–24], our work goes beyond previous findings in two important respects. First, previous studies have focused on a single facial dimension (e.g., identity[22–24]; viewpoint[21]), and hence could not address the relative timing of extraction of multiple dimensions of face perception. Generally, however, our onset latencies lie within the range of onset times reported in these studies. For example, prior studies found that the onset of viewpoint encoding across identity emerged at 60 ms after stimulus onset, while effects of viewpoint symmetry (supporting view-point invariant mechanisms of face identification) started at around 80 ms[21]. Further, another study reported identity decoding accuracy across changes in expression at around 50 ms[24], although that study used a 60ms-sliding temporal window hence this time point could reflect neural signals up to 110 ms after stimulus onset. To our knowledge, only one prior study[23] investigated the representation of identity within and across gender, and reported no difference in onset latencies (onset ~60 ms for each). Their analysis, however, likely included low-level stimulus differences in the identity and gender comparisons due to a limited set of stimuli, as well as identity information in the gender comparison, whereas we investigated gender and identity information unconfounded from each other and from low-level features (i.e., by partialling out the irrelevant models). Overall, while prior studies find similar early decoding of face identity information, consistent with our results,

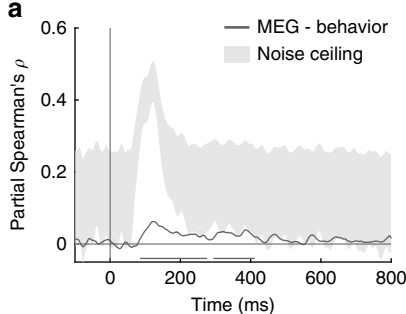
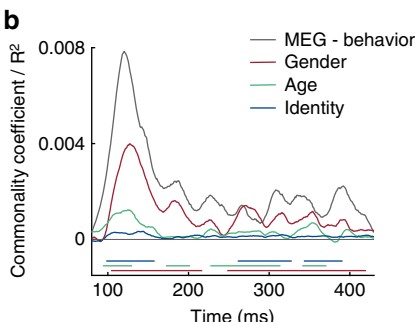

**Fig. 4** MEG and behavioral RDM comparisons. **a** Time course of correlation (partial Spearman, partialling out low-level features) between behavioral and MEG RDMs ($n = 14$). Gray-shaded area indicates estimated noise ceiling based on the variability across subjects. **b** Model-based commonality analysis showing the portion of shared variance between MEG and behavior (gray line) uniquely explained by gender (i.e., commonality coefficient; red), age (green) and identity (blue). Note that this analysis was restricted to the time window of the significant correlation between MEG and behavior (as shown in **a**). Colored lines below plots indicate significant times using cluster-based sign permutation test (cluster-defining threshold $p < 0.05$, and corrected significance level $p < 0.05$). Source data are provided as a Source Data file

they do not reveal the relative timing of extraction of different dimensions from faces.

Second, and most importantly, our study is the first to reveal how and when familiarity affects the representations of different face dimensions. The neural mechanisms underlying the powerful behavioral benefit for the perception of familiar versus unfamiliar faces[7,25] have remained an important unsolved mystery. In particular, is the familiarity enhancement effect due to tuning of bottom-up perceptual filters, or to top-down feedback[8,9,11]? Previous studies could not address this question as they used only unfamiliar face images[22–24], indirect analyses of steady-state visual evoked responses[26] or functional magnetic resonance imaging with low temporal resolution[27–31]. Our finding that identity and gender information are significantly enhanced for familiar faces, very early on in processing, and virtually as early as this information becomes available, suggests a neural mechanism for the strong behavioral enhancement in perception of familiar versus unfamiliar faces. In particular, because there is unlikely sufficient time for feedback from high-level areas[32], this early enhancement is most likely due to tuning of feed-forward processing for familiar faces or local recurrent processes[17]. Together, our results suggest that familiarity affects face perception by altering feed-forward face processing, not exclusively through feedback from later stages after the personal identity of a face has been extracted.

How do our findings relate to prior findings on the spatial organization of face processing in the brain? In our study, we consider whole-brain information and do not restrict the time course to certain spatial locations given the low spatial resolution of MEG. Despite the lack of spatial resolution, the finding that gender and age were extracted before identity information suggests that these aspects of face perception are processed at different stages of face processing. Evidence from fMRI is equivocal about where specific face dimensions such as gender[33,34] or identity[35–37] are represented in the brain. Given the fleeting presence of face dimensions in our MEG data, it is possible that fMRI misses some of this information due to its low temporal resolution. Interestingly, previous studies found that people with developmental prosopagnosia, who have trouble recognizing faces, have no impairments in gender processing, suggesting that these two facial dimensions might be processed in distinct neural areas, potentially in parallel[38,39]. In contrast, behavioral studies showed that gender processing influences face identification[6]. Our results reconcile these two findings by suggesting that gender and age are processed earlier in the processing hierarchy, at stages possibly less affected by prosopagnosia, yet able to influence subsequent identity processing in typical subjects[6]. Taken together, it remains unclear where gender, age or identity are represented in the brain. In future, this question might be answered by combining MEG with fMRI using a fusion approach[16,17] to link our finding to regions in the brain.

While human neuroimaging techniques are mainly limited to either high temporal or high spatial resolution, neurophysiological recordings in non-human primates provide an ideal opportunity to simultaneously measure face representations with high temporal and spatial resolution. Indeed, the macaque face perception system is similar[40,41] and possibly homologous[42] to the human face processing system. Neurophysiological studies with non-human primates find that categorical distinctions between faces and other categories develop earlier than face identity information[41,43], but it is still unknown when other face dimensions, such as gender or age, emerge. Consistent with human MEG data[21], facial representations in macaques were also found to gradually build up and become more invariant to viewpoint at successive processing stages, measured both spatially and temporally[40,44], again showing the usefulness of the macaque's face perception system as a model to study human face perception. With regard to familiarity, our findings are in line with a recent study in macaques reporting early quantitative differences and late qualitative differences in processing of familiar versus unfamiliar faces[45]. However, the paradigms and stimuli that have been used so far in humans and macaque studies are too different to provide a precise correspondence between species.

Our findings place important constraints on computational models of face perception and further suggest new hypotheses to probe in such models. Recently, deep convolutional neural networks (CNNs), have provided successful models of object and scene perception in humans and macaques[46–48]. However, it remains unclear how useful CNNs are as model for human face processing. In future work, it will be interesting to test how well different computational models can explain our findings. In particular, the fact that we found little temporal cross-decoding (see Supplementary Note 4) suggests that a sequence of non-linear operations are performed on the representations of each face dimension. Comparisons of our data to different computational models might shed light on the operations and transformations performed at different stages of face processing in humans, as has been successfully done for facial viewpoint decoding in macaques[49]. Furthermore, the face dimensions tested in this study did not yet reach the noise ceiling, suggesting that other factors beyond the ones investigated in this study might contribute to the processing of faces. In future, it will be important for computational models of face perception to determine the relevant dimensions for face processing. Most importantly, a crucial implication of our data is that early stages of face processing are apparently tuned to familiar faces, a phenomenon that could also be tested in computational models. Analyzing these questions could provide a path toward the development of a computationally precise, image-computable model of face processing in humans.

While the results presented here further our understanding of human face processing dynamics, they also have several limitations. First, our decision to use natural images in this study for ecological validity may introduce greater low-level image confounds compared to highly controlled, artificial face stimuli. Although we partialled out low-level features as measured by an early layer of a CNN trained on faces (a conservative choice, given that this model had the greatest overall correlation with the MEG data), we cannot be sure that all low-level features have been captured by this model. Second, by investigating the onset of extraction of several face dimensions, we cannot draw conclusions about when the processing is completed, because the earliest latency of significant decoding reflects an upper bound for the beginning of the process. In fact, the correlations with gender, age and identity were relatively sustained until at least 400 ms after stimulus onset. Third, our analysis of familiarity effects is based on a comparison of neural responses to American and German celebrity stimuli and any low-level differences between these stimulus sets could have contributed to these effects. Our analysis of an early and a late layer of VGG-Face (see Supplementary Note 4) found no evidence for systematic differences in the American and German face stimuli, but it is possible they exist and were not detected in this analysis. In future, it would be useful to replicate our effects in a cross-over design in which one group of subjects is familiar with half of the identities selected as stimuli, whereas a second group is familiar with the other half. Lastly, the mere existence of representations revealed by multivariate pattern analysis does not in itself imply that these representations are relevant to behavior[18,19]. Here, we correlated MEG to behavioral similarity of our subjects and found that all face dimensions explained unique variance between MEG and behavior. While

this is an important step towards linking MEG representations with behavior, more direct links such as correlates to online behavior during MEG recording would be useful. However, our results do make testable predictions about human face processing behavior. For example, in line with our results, studies investigating the speed of human categorization behavior have shown that identity decisions were made faster than familiarity decisions[1] and that gender decisions occurred faster for familiar than unfamiliar faces[50]. Our results go beyond these findings to predict that behavioral discriminations of gender (and age, if tested in a binary fashion) should be made faster than discriminations of identity.

In sum, our findings of how face processing unfolds over time in humans show that the extraction of face dimensions follows a coarse-to-fine time trajectory, and support the hypothesis that the face processing system is tuned to familiar face features in a bottom-up manner. These findings inform broader questions about how prior experience affects processing of other stimuli beyond faces, such as object shapes[12–14], offer powerful constraints on computational models of face perception, and provide new predictions to be tested in future work.

## Methods

**Participants**. Twenty-one healthy volunteers with normal or corrected-to-normal vision participated in the study. Five subjects were excluded before data analysis due to at least one of the following exclusion criteria: excessive motion during the recording, behavioral performance below two standard deviations of the mean, or incomplete recordings due to technical issues. Data from 16 subjects (eight female; mean age 25.9, SD = 4.33) remained for the MEG analysis. The chosen sample size was based on previous studies using multivariate decoding of EEG/MEG data[16,17,23]. Fourteen of these 16 subjects additionally participated in an online behavioral follow-up experiment. All subjects provided informed, written consent prior to the experiment. The Massachusetts Institute of Technology (MIT) Committee on the Use of Humans as Experimental Subjects approved the experimental protocol (COUHES No 1606622600) and the study was conducted in compliance with all relevant ethical regulations for work with human participants.

**Experimental design and stimuli**. To investigate the temporal dynamics of face processing, subjects viewed face images of different identities while monitoring for consecutive repetitions of identical images (i.e., 1-back task; Fig. 1a) in the MEG. We chose eight familiar (i.e., famous actors in the US) and eight unfamiliar (i.e., German actors) celebrities as identities, which varied orthogonally in gender and age, such that half were female and half were male and half of them were young (i.e., maximum age was 36 years) and half were old (i.e., minimum age was 59 years). Note that here, by gender, we refer to the sex of a face.

To ensure that all subjects were in fact familiar with the set of familiar identities, subjects completed an online screening task prior to the study. In this screening, we presented them with one image for each of the 16 identities (different from the images used in the MEG study) and asked if they were familiar with the person shown. Only subjects who recognized each of the eight familiar identities (e.g., by giving their names or contexts in which they remembered the person) were included in the study.

Final stimuli used in the MEG study consisted of five gray-scale images of each of the 16 identities for a total of 80 stimuli. For each identity, we selected five images from the internet which varied in several aspects such as expression (at least two smiling and two neutral facial expressions), eye gaze (one averted to the left, one averted to the right, two directed gaze and one gaze aligned with rotated head), pose (one with head slightly rotated to the side), lightning, hair, etc. We then standardized all images to a template by rotating, scaling and cropping them based on the position of the nose tip, the mouth center and both eyes and saved them as gray-scale images.

During the MEG experiment, subjects viewed trials of face images (Fig. 1a). Each trial started with the presentation of a face image for 0.2 s followed by a 0.8–1 s interstimulus interval (ISI; uniformly sampled between 0.8 and 1 s) during which a gray screen was presented. Subjects were instructed to respond via button press to a consecutive repetition of an identical image during image presentation or during ITI. To avoid artifacts due to eye movements or blinking, subjects were instructed to fixate a black fixation cross in the upper center of the screen during image presentation (i.e., presented between the tip of the nose and the eyes of a face) and ISI. They were further asked to blink at the same time when giving a button response, as these trials were not included in the data analysis.

Subjects viewed 28 blocks of trials in which each of the 80 images was presented once randomly interleaved with 20 task trials (1-back task) for a total of 100 trials per block. Task trials were pseudo-randomized such that each of the 80 images was additionally shown seven times as task trial for a total of 35 presentations. Stimulus

presentation was controlled and responses collected using Psychtoolbox 3 for Matlab[51,52]. The experiment lasted around 70 min.

**MEG recording and preprocessing**. MEG data were collected using a 306-channel Elekta Triux system with a 1000 Hz sampling rate, and were filtered online between 0.01 and 330 Hz. The position of the head was tracked during MEG recording based on a set of five head position indicator coils placed on particular landmarks on the head. We preprocessed the raw data with Maxfilter software (Elekta, Stockholm) to remove head motion and to denoise the data using spatiotemporal filters. We then used Brainstorm (version 3.4[53]) to extract trials from −200 to 800 ms with respect to image onset. In Brainstorm, every trial was baseline-corrected by removing the mean activation from each MEG sensor between −200 ms and stimulus onset and principal component analysis was used to remove eye blink artifacts which were automatically detected from frontal sensor MEG data. We used a 6000 fT peak-to-peak rejection threshold to discard bad trials, imported the remaining trials in Matlab (version 2016a; The Mathworks, Natick, MA) and smoothed them with a 30 Hz low-pass filter. Note that we also performed an analysis on the unfiltered data which yielded very similar results (see Supplementary Note 2). To further decrease noise and to reduce computational costs, for each subject we concatenated data of each MEG sensor over time and applied principal component analysis to the MEG sensor data (keeping all components that explained 99.99% of the variance in the data). This step reduced the set of features from 306 MEG sensors to around 70 principal components (PCs) per subject and we conducted all further analysis on this reduced set. We then baseline-corrected every trial by removing the mean activation between −200 ms and stimulus onset from each PC. These PC scores for each trial and each time point were used for the subsequent analyses.

**MEG multivariate pattern analysis**. We used multivariate pattern analysis to extract temporal information about the face stimuli from the MEG data (Fig. 2). To obtain a similarity measure for each pair of stimuli, we used cross-validated pairwise classification accuracy of linear support vector machines (SVM; libsvm[54]). Classification analysis was performed separately for each subject in a time-resolved manner (i.e., independently for each time point). A pattern in the analysis consisted of the PC scores for one trial and one condition at a given time point. In the first step, we sub-averaged all trials of one condition by randomly assigning each trial to one of five splits and averaging the trials in each split (~5–7 trials per split when considering bad trials). We then divided the groups into training and testing data randomly selecting one group for testing and the remaining groups for training (i.e., five-fold cross-validation). We then conducted a binary classification of all 3170 pairwise comparisons (i.e., 80 × 79/2 combinations) between conditions. This classification procedure was repeated 100 times. The average decoding accuracies over repetitions served as value in the 80 × 80 decoding matrix, termed representational dissimilarity matrix (RDM). This RDM is symmetric and the diagonal is undefined. The entire procedure resulted in one MEG RDM for each subject and time point.

To get a measure of how well each face stimulus can be discriminated from all other images in the MEG (i.e., image decoding), we averaged all pairwise decoding accuracies in the lower triangular of each RDM. This resulted in one average decoding accuracy value per subject and time point. The time course of image decoding further serves as benchmark of time course of low-level image processing in the MEG data. To investigate how persistent neural responses were to face images, we further extended the SVM decoding procedure with a temporal generalization approach[16,55,56]. Details and results of this analysis can be found in the Supplementary Note 4.

**Representational similarity analysis**. To analyze the representation of face dimensions in the MEG data, we used representational similarity analysis (RSA). We created model RDMs for each face dimension which were 80 × 80 binary matrices where 1 corresponded to a between category stimulus comparison (e.g., male vs female for the gender model) and 0 to a within category stimulus comparison (e.g., female vs. female). This procedure resulted in four face models corresponding to the familiarity, gender, age and identity dimensions of our stimuli. To compute correlations between each model and the MEG data, we extracted the lower off-diagonal of each of these matrixes as vectors. For each model and subject, we computed the partial rank coefficients (Spearman correlation) between the model and the MEG RDM at each time point partialling out all other face models. This step was crucial because some of the models are correlated (e.g., between identity comparisons comprised between gender comparisons) and partialling out the other models thus allowed us to disentangle contributions of the models from each other.

To further exclude the contribution of low-level features of our stimuli to the results, we additionally partialled out a low-level feature model. This low-level feature model was computed by extracting features for each of the 80 stimuli from the second convolutional layer of a deep, convolutional artificial neural network (CNN) trained on thousands of face identities (VGG-Face[57]). We used 1 − Pearson correlation as a measure of dissimilarity between the CNN units of each pair of stimuli, resulting in a 80 × 80 RDM based on low-level image features. Note that we also compared other models of low-level features (e.g., HMAX C2[58,59], Gist[60],

pixel-based similarity), which produced similar results; we report here the VGG-Face model because it reached the maximum correlation with the MEG data and hence explains the most data (as accountable by low-level features).

We investigated the effect of familiarity on face processing by dividing the MEG and model RDMs into within familiar and within unfamiliar RDMs, respectively. Each of these RDMs was a $40 \times 40$ RDM constituting of only familiar or only unfamiliar face images. We then performed the same analysis as for the full set of stimuli (see above). To further test differences between familiar and unfamiliar face processing, we subtracted the time courses of correlation for unfamiliar faces from the time courses obtained for familiar faces for each subject and statistically compared these difference time courses to zero (see Statistical inference below). Note that while we tried to select the different sets of familiar and unfamiliar face images as objectively as possible, we cannot fully exclude that differences between the sets of stimuli contributed to this analysis. We therefore performed an additional analysis of VGG-Face, testing for stimulus-driven familiarity effects in an early and a late layer of VGG-Face, suggesting that such differences could not straightforwardly explain our findings (see Supplementary Note 1).

Further, it is important to note that categorical information time series (e.g., gender) were constructed by correlating the MEG RDM matrix with model RMDs consisting of zeros corresponding to within-category (e.g., female or male) and ones corresponding to between-category stimulus comparisons. The correlation between the MEG RDMs and a model RDM (while partialling out all other models) served as a measure of clustering by category membership. An alternative approach to computing categorical information time series is to directly train a classifier to discriminate categories (e.g., female versus male across identity) stimuli. While such a methodological approach may be sensitive to different aspects of categorical stimulus information in general, it yielded consistent results in our data (see Supplementary Note 3).

**Behavioral similarity experiment**. Fourteen of the 16 subjects additionally performed a behavioral multi-arrangement task[61] on the same stimuli on a separate day after the MEG experiment. Subjects performed the multi-arrangement experiment online using their own computer and by logging into an online platform to run behavioral experiments ([www.meadows-research.com]). Subjects had to enter an anonymous, personal code that was provided to them via email to start the experiment. In the experiment, all 80 stimuli that the subject had previously seen in the experiment were arranged as thumbnails around a white circle in the center of the screen. Subjects were instructed to arrange these thumbnails based on their perceived similarity ("similar images together, dissimilar images apart", without explicit instructions on which feature to use) by dragging and dropping them in the circle. The experiment terminated automatically when a sufficient signal to noise ratio was reached (i.e., evidence weight was set to 0.5). The average duration of the experiment was ~70 min. After the completion of the experiment, the pairwise squared on-screen distances between the arranged thumbnails was computed, thus representing a behavioral RDM. For each subject, we extracted the lower off-diagonal data from the behavioral RDM and correlated this vector with the corresponding MEG RDMs for each time point. We additionally computed the noise ceiling for this correlation to get an estimate for the upper and lower bound of the correlation given the variability across the restricted set of subjects in this analysis. We estimated the noise ceiling following a method described here[62]. Briefly, we estimated the upper bound of the correlation as the mean correlation of each subject with the group mean. As this correlation includes the correlation with the subject itself, it represents an overestimation of the true model's average correlation. In contrast, the lower bound is computed by taking the mean correlation of each subject with the mean of all other subjects (excluding the subject itself). This underestimates the true model's average correlation due to restricted set of data. Together, the noise ceiling provides an estimate of the maximum obtainable correlation and is useful as a reference, in particular when low but significant correlation values are found.

Further, to assess the unique contribution of each model to the shared variance between MEG and behavioral RDMs, we additionally performed commonality analysis, a variance partitioning approach that estimates the shared variance between more than two variables[20,63]. Briefly, we computed the variance uniquely contributed from each face model (e.g., gender) by calculating two correlation coefficients: First, for each subject, we calculated the partial correlation between MEG and behavioral RDMs, while partialling out all models (gender, age, identity and low-level feature model). Second, we calculated the partial correlation between MEG RDM and behavioral RDM while partialling out all face models and the low-level feature model but leaving one face model out (e.g., gender). The difference between these two partial correlation coefficients represents the unique variance contributed by that model referred to as commonality coefficient. This step was repeated for every MEG time point resulting in a commonality coefficient time course for each face model.

**Statistical inference**. For all analyses, we used non-parametric statistical tests that do not rely on assumptions on the distributions of the data[64,65]. For statistical inference of decoding accuracy (image decoding) or partial correlation (e.g., model correlation) time series, we performed permutation-based cluster-size inference (i.e., a cluster refers to a set of contiguous time points). The null hypothesis corresponded to 50% chance level for decoding accuracies, and 0 for correlation

values or correlation differences. Significant temporal clusters were defined as follows. First, we permuted the condition labels of the MEG data by randomly multiplying subject responses by $+1$ or $-1$ (i.e., sign permutation test). We repeated this procedure 1000 times resulting in a permutation distribution for every time point. Second, time points that exceeded the 95th percentile of the permutation distribution served as cluster-inducing time points (i.e., equivalent to $p < 0.05$; one-sided). Lastly, clusters in time were defined as the 95th percentile of the maximum number of contiguous, significant time points across all permutations (i.e., equivalent to $p < 0.05$; one-sided).

**Onset and peak latency analysis**. To test for statistical differences in onset or peak latencies between different face dimensions, we performed bootstrap tests. We bootstrapped the subject-specific time courses (e.g., measured as decoding accuracy, partial correlation or commonality coefficient) 1000 times to obtain an empirical distribution of the onset (i.e., minimum significant time point post stimulus onset) and peak latencies (i.e., maximum correlation value between 80 and 180 ms post stimulus onset). We restricted the time window for the peak analysis to 180 ms post stimulus onset, since we were interested in the first peak occurring after stimulus onset, unconfounded from later peaks (e.g., due to stimulus offset responses[66]). The 2.5th and the 97.5th percentile of these distributions defined the 95% confidence interval for onset and peak latency, respectively. For differences between latencies, we computed 1000 bootstrap samples of the difference between two latencies (e.g., onset) resulting in an empirical distribution of latency differences. The number of differences that were smaller or larger than zero divided by the number of permutations defined the p-value (i.e., two-sided testing). These p-values were corrected for multiple comparisons using false discovery rate (FDR) at a 0.05 level.

## Data availability
The stimuli used in this study can be downloaded from the Open Science Framework ([https://osf.io/gk6f5/]). MEG data will be made available upon request. The VGG-Face model is available online ([http://www.robots.ox.ac.uk/~vgg/software/vgg_face/]). We used Psychtoolbox 3 ([www.psychtoolbox.org]), Elekta MaxFilter software, Brainstorm ([https://neuroimage.usc.edu/brainstorm]), Meadows ([www.meadows-research.com]), the libsvm toolbox and standard Matlab (R2017b) functions for data collection and analysis. The source data underlying Figs. 2, 3 and 4 and Supplementary Figs. 1, 2, 3 and 4 are provided as a Source Data file.

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

## Acknowledgements

This work was supported by a Feodor-Lynen postdoctoral fellowship of the Humboldt Foundation to K.D., NIH grant Grant DP1HD091947 to N.K and National Science Foundation Science and Technology Center for Brains, Minds, and Machines.

## Author contributions

K.D., L.I. and N.K. designed the experiment. K.D. collected the data. D.P. and L.I. provided code for data analysis. K.D. analyzed the data. K.D., L.I., D.P. and N.K. wrote the article.

## Additional information

**Competing interests:** The authors declare no competing interests.

