## [Peer Review File · Nature Communications]

Reviewers' comments:

Reviewer #1 (Remarks to the Author):

The study addresses the neural dynamics of face processing as revealed by multivariate analyses as applied to MEG data. In contrast to previous work, the study examines and compares multiple aspects of face recognition including identity, gender and age as well as familiarity. The results of the study are interpreted as evidence for a coarse-to-fine mode of visual processing and for tuning of early feed-forward mechanisms of familiar face recognition.

The topic is of general interest and the results are in overall agreement with prior work. Also, the logic of the study is straightforward and the manuscript is clearly written. Some important results are missing from the current manuscript though and some of the conclusions could benefit from further discussion.

1. What is the decoding accuracy for identity, gender, age and how do they differ from each other? Only the results of image decoding are shown (Fig 2a). The time course of correlation is informative but that is a subsequent analysis relying on the success of decoding. Also, it would be interesting to examine whether the time course of decoding mirrors that of correlation-based analyses.

2. Is classifier training and testing (for identity, gender, age) carried out across the same stimuli or across different ones? Training and testing across different but homologous stimuli (e.g., by training the classifier on two different individuals of different genders and then testing on two other individuals of different genders) is commonly used to rule out the contribution of irrelevant factors to decoding. In the present case, it could serve as a more direct approach of handling irrelevant image differences compared to partialling out model-based similarity estimates (to which the authors dedicate considerable attention).

3. All correlation values (Fig 2b) are relatively small (<0.1). What factors might account for the unexplained structure of the data (other than noise)?

4. The authors interpret early familiarity effects as evidence for the role played by familiarity in feed-forward visual processing. However, the possibility of fast feedback processing is also raised (p. 7). Please discuss the relative strength/merit of these different hypotheses.

5. Early familiarity effects (Fig 3d) rely on comparison with chance-level correlation values for unfamiliar faces. This would recommend caution in the interpretation of the results. On a related note, is decoding accuracy for unfamiliar faces significantly above chance (please see also point 1 above)? If not, this could be problematic since classification did not manage to identify data structure relevant for unfamiliar faces and their corresponding identity representations.

6. It is interesting that correlations with behavioral data peak relatively early (around 123 ms). Does this suggest that the behavioral task relies primarily on early low-level visual processing?

7. Please describe how exactly noise ceiling (Fig 4a) was computed and how its results are informative.

Reviewer #2 (Remarks to the Author):

The authors use MEG and representational similarity analysis to investigate the neural time course of face processing, focusing on the ability to distinguish the gender, age and identity of faces. A few recent studies have used EEG or MEG to investigate the neural discrimination of face identity or face viewpoint in a time-resolved manner but these previous studies did not investigate effects of gender or age as separate variables or test for effects of face familiarity. The reported findings of this study include some novel components that should of strong interest to face perception researchers.

While the experimental methods and results are quite strong, a few aspects of the study are puzzling. First, the earliest time point at which gender and age can be distinguished is unusually early, 60 ms and 72ms, respectively. This is much earlier than the times reported in other recent

studies for discriminating face viewpoint or face identity across changes in expression, which found discriminating effects to emerge at around 100ms. One methodological concern is that the authors have applied a low-pass temporal filter to their data. This could potentially shift the time of earliest discrimination to an earlier point in time, by effectively shifting/blurring information in the temporal domain for the filtered MEG signals. What happens if the same analysis is performed on unfiltered data?

Another concern is that the later discrimination of face identity, as compared to discrimination of face gender or age, might simply be due to lower statistical power at detecting these more subtle effects. It is not clear in Figure 2B that the peak in discriminating face identity occurs later in time than the effect of gender or age. If all of the relevant face discriminating signals tend to emerge at a similar time, then the present findings would largely agree with other recent findings.

The reported effect of better discrimination of face gender for familiar faces is interesting but controls are needed to show that this effect cannot be attributed to stimulus differences between the American and German celebrity faces. What if the American faces are more sexually dimorphic than the German faces? Can some type of control analysis be performed to address this issue? If participants are given a speeded gender classification task, is performance comparable for American and/or German participants? Or might the analysis of the VGG-Face model be used to address concerns of potential differences in the faces used for these two stimulus sets?

Minor comments

The statement about the lack of temporal stability of the neural representations seems inaccurate in its current framing. EEG and MEG signals fluctuate over space/time, and the lack of stability could be due to the instability of the relevant signal itself or due to the instability of noisy or irrelevant signals, such that the spatial pattern of the relevant signal does not appear consistent over time.

Likewise, the statement about image-invariant identity information may need to be tempered. Would a low-level classifier, such as one trained on the second layer of the CNN, perform at chance level at generalizing across different images of same identity?

We thank both reviewers for their thoughtful and helpful comments on our manuscript. We have addressed all the comments in the revised manuscript, as detailed below. We have also added several new analyses. For example, to test the possible influence of temporal filtering on the onset latencies, we have conducted an additional analysis on the unfiltered data (now included in the Supplementary Information). This analysis confirms the pattern of latencies we observed for the onset of gender, age, and identity information in our original analysis. Second, we have tested potential stimulus differences between familiar and unfamiliar face images with a new analysis of the representations of these images in VGG-Face, as suggested by one of the reviewers. This new analysis shows that stimulus differences cannot straightforwardly explain our MEG results. Finally, as suggested by one reviewer, we added a complementary cross-decoding analysis (e.g., train gender across identities, test gender on left-out identities). This new analysis shows a time course of cross-decoding accuracy that confirms the results from the RSA analysis we had conducted previously, and has been added to the Supplementary Information. We have further changed the discussion and clarified the methods to address points raised by the reviewers. All of these changes are detailed below and highlighted in the revised manuscript. We believe these changes have led to a substantially improved manuscript.

Reviewer #1 (Remarks to the Author):

The study addresses the neural dynamics of face processing as revealed by multivariate analyses as applied to MEG data. In contrast to previous work, the study examines and compares multiple aspects of face recognition including identity, gender and age as well as familiarity. The results of the study are interpreted as evidence for a coarse-to-fine mode of visual processing and for tuning of early feed-forward mechanisms of familiar face recognition. The topic is of general interest and the results are in overall agreement with prior work. Also, the logic of the study is straightforward and the manuscript is clearly written. Some important results are missing from the current manuscript though and some of the conclusions could benefit from further discussion.

1. What is the decoding accuracy for identity, gender, age and how do they differ from each other? Only the results of image decoding are shown (Fig 2a). The time course of correlation is informative but that is a subsequent analysis relying on the success of decoding. Also, it would be interesting to examine whether the time course of decoding mirrors that of correlation-based analyses.

We apologize that this was unclear from the description of our methods. As the reviewer notes, an alternative method to representational similarity analysis (RSA) we used for measuring the time course of information extraction is to calculate classifier accuracy for each dimension at each timepoint after stimulus onset. We agree with the reviewer that category classifier accuracy measures are useful, so we have performed a new cross-decoding analysis. This analysis yielded highly similar results, and we added this analysis to the Supplementary Information (see also point 2 below), along with a more detailed explanation of the differences between the two approaches to the Methods, section "Representational similarity analysis" (p. 17).

However, we have retained our original RSA approach as the main analysis in the paper for several reasons outlined below. To first better explain what this analysis does: It computes the correlation between MEG RDMs (based on pairwise image decoding accuracies) at each timepoint with model RDMs representing gender, age, and identity information (see Fig. 2b). In

more detail: Our analysis starts by calculating an MEG RDM, which is a pairwise image decoding matrix from the MEG responses at each timepoint. Each cell in the matrix of that RDM shows the accuracy of a classifier in decoding the MEG responses to the pair of images corresponding to that row and column of the matrix. To measure gender information (for example) at each timepoint, the MEG RDM for that timepoint is correlated with a model RDM reflecting binary gender classification. This gender RDM is a matrix of the same stimulus image pairs in which each cell has a zero if those two images are of the same gender and a 1 if those two images are of different gender. If the MEG RDM at a given timepoint contains gender information then it will be correlated with the gender model RDM. Similarly, an identity RDM is a matrix with a zero at each cell corresponding to two images of the same individual, and a 1 for any cell corresponding to two images of different people. By correlating the MEG RDM at each timepoint with model RDMs for gender, identity, and age, we get a measure of the amount of information about gender, identity, and age in the MEG signal at each timepoint. This RDM-based analysis method is quite standard, and has been widely used in the analysis of EEG/MEG data in previous studies (e.g. Dima et al., 2018; Grootswagers et al., 2017; Hebart et al., 2018; Kietzman et al., 2018; Vida et al., 2017).

Our reasons for choosing RSA, rather than using category classification accuracy as our main analysis in the paper, are multi-fold. First, we are interested in comparing data from different modalities (e.g. behavior and representations from VGG-Face). The computation of MEG RDMs, model RDMs and behavioral RDMs, allows us to compare between modalities, while controlling for other models. Second, the correlation approach allows us to compute a noise ceiling, thereby providing a measure of how much each face dimension contributes to the explainable variance in the data (see also point 3 below). In contrast, corresponding decoding accuracies do not provide this information. Third, the RSA approach used here enables us to directly compare gender, age, and identity information with each other using the same analysis scheme. A comparable direct decoding approach would require different classifiers with different training and testing schemes (e.g. leave-one-identity out for gender, leave-one image out for identity), with different statistical power for each. These classifiers cannot be trivially compared to each other. For all these reasons, we have retained the RSA analysis as our main analysis in the paper, while adding the classification accuracy analyses to the Supplemental sections.

2. Is classifier training and testing (for identity, gender, age) carried out across the same stimuli or across different ones? Training and testing across different but homologous stimuli (e.g., by training the classifier on two different individuals of different genders and then testing on two other individuals of different genders) is commonly used to rule out the contribution of irrelevant factors to decoding. In the present case, it could serve as a more direct approach of handling irrelevant image differences compared to partialling out model-based similarity estimates (to which the authors dedicate considerable attention).

We agree with the reviewer that direct cross-decoding of categorical dimensions should similarly control for these other dimensions (but see our response to the last point of Reviewer 2 where this is not the case). We have therefore now computed cross-decoding accuracies for all face dimensions, added this analysis to the Supplementary Information, and referred to it in the Results (p. 6) and Methods sections (p.17). For gender, age, and familiarity, these decoding accuracies are across identities (e.g. leave-one-identity out), and for identity the decoding accuracy is across images (e.g. train on all but one image of a person and test on the left-out image of that person). As can be seen in Figure S3, the time course of cross-decoding largely replicates the findings in Figure 2b.

3. All correlation values (Fig 2b) are relatively small (<0.1). What factors might account for the unexplained structure of the data (other than noise)?

We agree with the reviewer that the obtained correlation values are small. However, please note that our correlation values are in line with other similar studies (e.g. Dima et al., 2018; Grootswagers et al., 2017; Hebart et al., 2018; Vida et al., 2017). As mentioned by the reviewer, one limiting factor is noise in the data. A common way to compute an estimate of the noise introduced through variability across subjects has been proposed by Nili et al. (2014). The authors suggest computing the average correlation between each subject's MEG data and the group mean (including the current subject) as an upper bound of the intersubject noise ceiling, and the average correlation between each subject's data and the mean of all other subjects (excluding the current subject) as a lower bound of the intersubject noise ceiling. We calculated the intersubject noise ceiling for the data based on all 16 subjects and obtained a noise ceiling of around 0.48 (max. lower bound: 0.43, max. upper bound 0.52) which limits the maximum correlation that can be obtained. Further, it is important to note that the correlations reported here are the unique contributions of each face dimension to the MEG data. By partialling all other models out, the correlations are significantly reduced.

Another important factor is the existence of other non face-specific image features that are processed by the visual pathway and presumably manifested in the observed MEG response. To show how much such simple features can explain, we below show the correlation with the low-level feature model (i.e. the second convolutional layer of VGG-Face), as well as all other models compared to the noise ceiling. We agree with the reviewer that it will be interesting to explore the still unexplained variance and we hope that follow-up studies will shed more light on which other face-specific (e.g. friendliness, attractiveness of a face) or non-face-specific (e.g. background, curviness) dimensions further contribute to the neural processing of faces. We further added this interesting aspect of our data to the discussion (p. 12).

4. The authors interpret early familiarity effects as evidence for the role played by familiarity in feed-forward visual processing. However, the possibility of fast feedback processing is also raised (p. 7). Please discuss the relative strength/merit of these different hypotheses.

We thank the reviewer for pointing out this issue. We noticed that the phrase “fast feedback processes” was confusing as it suggested two different feedback hypotheses. In fact, by fast feedback (perhaps via local recurrent processes) we wanted to suggest a possible mechanism underlying feedforward enhancement. To clarify, we changed the sentence to: “Overall, familiarity enhancement arose early during processing, suggesting that early stages of visual processing are tuned to familiar face features (perhaps via local recurrent processes ¹⁷)” (p. 7). We further extended the discussion of this finding (p. 11).

5. Early familiarity effects (Fig 3d) rely on comparison with chance-level correlation values for unfamiliar faces. This would recommend caution in the interpretation of the results. On a related note, is decoding accuracy for unfamiliar faces significantly above chance (please see also point 1 above)? If not, this could be problematic since classification did not manage to identify data structure relevant for unfamiliar faces and their corresponding identity representations.

As outlined above (point 1 and 2), we used a pairwise image classifier to obtain MEG RDMS and correlated these with model RDMS. In case of the familiarity effect for identity, that means that for familiar faces, the MEG responses to two different images of the same person were more similar to each other than MEG responses to two images of different people. In contrast, this was not (or only to a small degree) the case for unfamiliar faces. So yes, we agree that we could not extract categorical information about identity from unfamiliar face images. The finding that familiarity enhances identity processing, however, is based on whether the difference between familiar and unfamiliar face processing is significant. This is a valid way to test an enhancement of identity processing. If anything, the lack of any significant identity discrimination from unfamiliar faces strengthens this result – the effect of familiarity is so strong that there is no detectable identity information for unfamiliar faces!

If the reviewer was concerned that the MEG representations failed to discriminate images (rather than identities) for unfamiliar faces (which would then also lead to chance-level correlations with the identity model), we show in the figure below the mean pairwise decodability separated for familiar and unfamiliar images. As can be seen, we did not find a significant difference between the decoding accuracy of familiar versus unfamiliar *images*. Importantly, while it could still be possible that higher-level differences between stimuli might have contributed to the familiarity enhancement effect, our analysis of a late VGG-Face layer does not support this hypothesis (see Supplementary Analyses and our response to Reviewer 2). Together, our results show that identity information is enhanced for familiar faces, in line with the strong familiarity enhancement effect previously found in behavior.

6. It is interesting that correlations with behavioral data peak relatively early (around 123 ms). Does this suggest that the behavioral task relies primarily on early low-level visual processing?

We thank the reviewer for raising this interesting question. While the peak for the correlation with behavior is early, it actually overlaps with the peak of the other face dimensions reported in Figure 2b and 2d which are also found around 125ms. We added this information to the results (p. 9). In contrast, the peak of low-level image processing, as shown for image decoding occurs around 100ms. Our model-based commonality analysis further suggests that the face dimensions gender, age and identity contribute to this correlation between MEG and behavior. We therefore believe that low-level features do not drive the correlation between MEG and behavior. However, as the latter analysis partials out the low-level features, it is still a valid point to ask whether or how much the subjects used low-level features to perform the behavioral task. To find out, we correlated the behavioral RDMs with all of our face models (while partialling out all other models). We attached the results of this analysis below. In line with our hypothesis, we find that subjects' behavior can be best explained by gender, age and identity models, whereas familiarity and low-level features hardly correlate with behavior.

7. Please describe how exactly noise ceiling (Fig 4a) was computed and how its results are informative.

We thank the reviewer for pointing out the missing description of the noise ceiling in our manuscript. We have now added the motivation and description for this analysis to the Methods in the section “Behavioral similarity experiment” (p. 18).

Reviewer #2 (Remarks to the Author):

The authors use MEG and representational similarity analysis to investigate the neural time course of face processing, focusing on the ability to distinguish the gender, age and identity of faces. A few recent studies have used EEG or MEG to investigate the neural discrimination of face identity or face viewpoint in a time-resolved manner but these previous studies did not investigate effects of gender or age as separate variables or test for effects of face familiarity. The reported findings of this study include some novel components that should of strong interest to face perception researchers.

While the experimental methods and results are quite strong, a few aspects of the study are puzzling. First, the earliest time point at which gender and age can be distinguished is unusually early, 60 ms and 72ms, respectively. This is much earlier than the times reported in other recent studies for discriminating face viewpoint or face identity across changes in expression, which found discriminating effects to emerge at around 100ms. One methodological concern is that the authors have applied a low-pass temporal filter to their data. This could potentially shift the time of earliest discrimination to an earlier point in time, by effectively shifting/blurring information in the temporal domain for the filtered MEG signals. What happens if the same analysis is performed on unfiltered data?

We thank the reviewer for pointing out this valid concern. We agree that the use of a low-pass filter might have affected the onset latencies reported here. To address this concern, as suggested by the reviewer, we performed an additional analysis using the unfiltered data and added this analysis to the Supplementary Analyses and refer to it in the Methods (p. 15). This analysis largely mirrored the findings for the filtered data (see Fig. S2 in the Supplementary Analyses). While these results support the onset latencies we originally reported, we would further like to mention that our findings are within in the range of onset times based on similar analyses reported in other studies. For example, Kietzmann et al. (2017, p. 645) reported that the onset of viewpoint encoding across identity emerged at 60 ms after stimulus onset, while effects of viewpoint symmetry (a view-point invariant mechanisms of face identification) started at around 80 ms. Vida et al. (2017, p. 389) further reported identity decoding accuracy across changes in expression starting at around 50 ms (note however that they used a 60ms-sliding window which could potentially blur the onset up to 110 ms after stimulus onset). Overall, we conclude that the onsets reported here are unlikely due to artifacts of our data preprocessing, but rather represent the earliest onset of extraction of information given our stimuli.

Another concern is that the later discrimination of face identity, as compared to discrimination of face gender or age, might simply be due to lower statistical power at detecting these more subtle effects. It is not clear in Figure 2B that the peak in discriminating face identity occurs later in time than the effect of gender or age. If all of the relevant face discriminating signals tend to emerge at a similar time, then then present findings would largely agree with other recent

findings.

The reviewer raises a good point: In principle, the smaller effect magnitude of identity information than age or gender information could delay the onset of the significance of the identity information (relative to gender and age) simply because of the lower statistical power to detect the smaller identity effect. However, further analyses argue against this hypothesis. Specifically, identity information is almost three times as large for familiar faces analyzed alone (Fig. 3d; mean peak Spearman's r : 0.051) as it is for the full analysis across familiar and unfamiliar identities (Fig. 2b; mean peak Spearman's r : 0.018), even reaching a similar peak value to age information in the full analysis (Fig. 2b; mean peak Spearman's r : 0.057). Yet the onset latency is not affected by this increase in power. In fact, the onset latency of identity information even shows a trend in the opposite direction from 96 to 91 ms after stimulus onset for the familiar (high power) and full (low power) analysis, respectively. These data argue against the hypothesis that the later decoding of identity reflects lower statistical power. However, we take the reviewer's concern seriously and now discuss this potential issue in the manuscript (p. 6, p. 7).

Further, the reviewer is correct that the peak latencies do not differ across face dimensions. However we note that peak latencies are highly variable across subjects making this analysis very noisy, as found in other MEG studies (e.g., Isik et al., 2018). And, most importantly, our main interest is not when the presence of information is maximal, but when that information first appears in processing.

Concerning the reviewer's comment that our findings might largely agree with other recent findings, we would like to add that to our knowledge the only study comparing two (gender and identity) of the dimensions tested here is Nemrodov et al., 2016. While that study did not find a significant difference between gender and identity decoding onset latencies, our study addresses several limitations in their design: First, while their analysis is based on only 4 identities (2 female, 2 male) and 2 images per identity (see figure below), our stimuli are much richer by including many more identities and image variations. Second, the authors did not test the crucial comparison between gender and identity conditions and image decoding, while we showed that all our results are significantly later than image decoding. Further, and most importantly, our most novel result is the enhancement of information for familiar compared to unfamiliar faces, a question that has not even been addressed in prior MEG/EEG studies. Together, our findings go substantially beyond the findings reported in prior studies. We have revised the discussion to better reflect these advancements (p. 10).

The reported effect of better discrimination of face gender for familiar faces is interesting but controls are needed to show that this effect cannot be attributed to stimulus differences between the American and German celebrity faces. What if the American faces are more sexually dimorphic than the German faces? Can some type of control analysis be performed to address this issue? If participants are given a speeded gender classification task, is performance comparable for American and/or German participants? Or might the analysis of the VGG-Face model be used to address concerns of potential differences in the faces used for these two stimulus sets?

The reviewer raises a very good question. While we partialled out potential low-level differences between the two set of face images in our main analysis, we did not yet control for the possibility that high-level differences in the discriminability of gender or identity might contribute to our findings. While conducting an experiment with two groups of subjects who are either familiar or unfamiliar with each of the two sets would be an ideal control analysis, it is almost impossible to

conduct as almost every German person will at least be familiar with some of the US celebrities (while this was not the case vice versa). However, we agree with the reviewer that VGG-Face offers a great opportunity to test for purely stimulus-driven familiarity effects even on a higher level. We therefore performed an analysis in which we tested gender, age and identity information in an early and a late layer of VGG-Face, and added this analysis to the Supplementary Analyses and refer to it in the manuscript (p. 13, p. 17). This analysis indeed reveals two differences in age and identity discriminability of the German versus American faces in a late layer of VGG-Face. However, these differences cannot explain the results found in the MEG data: First, age shows higher discriminability for familiar than unfamiliar faces, an effect we did not find for MEG. Second, identity shows a small but significant advantage for unfamiliar over familiar identities which is the opposite of the familiarity enhancement effect we found for MEG. We therefore conclude that stimulus differences in the discriminability of the here tested face dimensions cannot straightforwardly explain our findings.

Minor comments

The statement about the lack of temporal stability of the neural representations seems inaccurate in its current framing. EEG and MEG signals fluctuate over space/time, and the lack of stability could be due to the instability of the relevant signal itself or due to the instability of noisy or irrelevant signals, such that the spatial pattern of the relevant signal does not appear consistent over time.

We agree with the reviewer that in principle lack of stability could be due to variable noise rather than variable signals. However, this seems unlikely for the following reasons. First, the temporal generalization analysis is a well-established technique used in multiple articles (e.g. Cichy et al., 2014; Mohsenzadeh, 2018; see King & Dehaene, 2014 for a review with several examples). Second, recent findings of this analysis on MEG data indeed found sustained neural stability, in which MEG representations for the orientation of the contrast edges were stable throughout the 800 ms stimulus presentation (see Fig. 4 in Pantazis et al., Neuroimage, 2018). Overall, the temporal generalization method works well in finding all ranges (from extremely transient to extremely stable) of representations in multiple studies. As these findings cannot exclude the possibility of variable noise in our data, we have added this alternative explanation to the Supplementary Analyses.

Likewise, the statement about image-invariant identity information may need to be tempered. Would a low-level classifier, such as one trained on the second layer of the CNN, perform at chance level at generalizing across different images of same identity?

The reviewer is raises a valid point. Our RSA analysis is only indirectly testing whether identity information can be generalized across images by testing whether neural representations of images within one identity (across images) are more similar than neural representation of images between identities. While it is possible, as suggested by the reviewer, that there is still sufficient low-level information about an identity even across images, we used the phrase “image-invariant identity information” based on the rationale that we controlled for this possibility by partialling out low-level features from our analysis. However, we agree with the reviewer that this term is too strong, and thus changed the sentence in which we previously used the term “image-invariant” to “This finding indicates that a late signature of generic familiarity can be read out from MEG signals, long after the onset of extraction of identity information” (p. 6).

We agree with the reviewer that it is an interesting question whether a classifier based on low-level features across images would also perform above chance level, in particular with respect to the cross-decoding analysis we additionally performed (see comments 1 and 2 to Reviewer 1). Is there still sufficient low-level information about an identity even across images that could be used by a classifier? To find out, we trained and tested a classifier to discriminate identity information across images (on pairs of identities with the same gender and age) using the features extracted from the second layer of VGG-Face. This classifier resulted in a mean decoding accuracy of 55.3% which reached significance ($p < 0.001$; two-sided permutation test). Similarly, the corresponding RSA analysis on the second layer of VGG-Face (i.e. correlating the identity model RDM with the RDM obtained from the second layer of VGG-Face) showed a significant effect (Spearman's r : 0.021; $p < 0.05$; two-sided bootstrap test). This finding reveals that in fact low-level information does exist that could support image-invariant identity decoding. That is why it is important to use an RSA approach (for capturing age, gender and familiarity effects) instead of using a cross-decoding approach: While the results of cross-decoding may still contain low-level information that is preserved across images within an identity, the RSA approach enables us to partial out low-level features in order to control for this irrelevant information. We further discuss the difference between these two analyses in the corresponding section of the cross-decoding analyses in the Supplementary Analyses.

REVIEWERS' COMMENTS:

Reviewer #1 (Remarks to the Author):

The authors have addressed my concerns.

Reviewer #2 (Remarks to the Author):

The authors have been thoughtful and very responsive in their revisions. They have performed several control analyses to address concerns that were raised in the previous review. This included performance of the same analyses on temporally unfiltered data to determine time of onset of reliable information, and an informative control analysis using VGG-Face to evaluate the visual discriminability of the American faces and German faces. The results of these control analyses bolster the findings and claims made in this study, and help strengthen the claims that familiarity modulates neural face processing.

One remaining issue is that the discussion section provides minimal description of the relationship between the current findings and previous findings using EEG or MEG. In particular, the latency at which different face processes emerge, and the findings of previous studies [Refs 21-24] would benefit from greater discussion in the main text. There are at least two short paragraphs in the supplement that are designed to address these issues. Moving much of this content into the main text is strongly recommended, so the reader can directly compare the reported latencies of various studies and make note of any differences reported across studies. Otherwise, the manuscript is in excellent condition and addresses all of my concerns.

We thank both reviewers for reviewing our manuscript. Their constructive comments and suggestions have substantially improved the manuscript. We have addressed the last minor issue raised by reviewer 2 below and in the final revision of the manuscript.

Reviewer #1 (Remarks to the Author):

The authors have addressed my concerns.

We are glad that we were able to address all concerns raised by the reviewer.

Reviewer #2 (Remarks to the Author):

The authors have been thoughtful and very responsive in their revisions. They have performed several control analyses to address concerns that were raised in the previous review. This included performance of the same analyses on temporally unfiltered data to determine time of onset of reliable information, and an informative control analysis using VGG-Face to evaluate the visual discriminability of the American faces and German faces. The results of these control analyses bolster the findings and claims made in this study, and help strengthen the claims that familiarity modulates neural face processing.

One remaining issue is that the discussion section provides minimal description of the relationship between the current findings and previous findings using EEG or MEG. In particular, the latency at which different face processes emerge, and the findings of previous studies [Refs 21-24] would benefit from greater discussion in the main text. There are at least two short paragraphs in the supplement that are designed to address these issues. Moving much of this content into the main text is strongly recommended, so the reader can directly compare the reported latencies of various studies and make note of any differences reported across studies. Otherwise, the manuscript is in excellent condition and addresses all of my concerns.

We thank the reviewer for this suggestion. We added a section discussing the latencies in comparison to previous studies to the discussion (p.10).